# Noli Me Tangere: Social Touch, Tactile Defensiveness, and Communication in Neurodevelopmental Disorders

**DOI:** 10.3390/brainsci9120368

**Published:** 2019-12-12

**Authors:** Daniela Smirni, Pietro Smirni, Marco Carotenuto, Lucia Parisi, Giuseppe Quatrosi, Michele Roccella

**Affiliations:** 1Department of Psychology, Educational Science and Human Movement, University of Palermo, 90128 Palermo, Italy; lucia.parisi@unipa.it (L.P.); michele.roccella@unipa.it (M.R.); 2Department of Educational Sciences, University of Catania, 95124 Catania, Italy; pietrosmirni@hotmail.it; 3Clinic of Child and Adolescent Neuropsychiatry, Department of Mental Health, Physical and Preventive Medicine, Università degli Studi della Campania “Luigi Vanvitelli”, 81100 Caserta, Italy; marco.carotenuto@unicampania.it; 4Dipartimento Promozione della Salute, Materno-Infantile, di Medicina Interna e Specialistica di Eccellenza “G. D’Alessandro”, 90127 Palermo, Italy; peppe.quatrosi@gmail.com

**Keywords:** tactile defensiveness, neurodevelopmental disorders (NDDs), sensory reactivity, autism spectrum disorders, social touch

## Abstract

Tactile defensiveness is a common feature in neurodevelopmental disorders (NDDs). Since the first studies, tactile defensiveness has been described as the result of an abnormal response to sensory stimulation. Moreover, it has been studied how the tactile system is closely linked to socio-communicative development and how the interoceptive sensory system supports both a discriminating touch and an affective touch. Therefore, several neurophysiological studies have been conducted to investigate the neurobiological basis of the development and functioning of the tactile system for a better understanding of the tactile defensiveness behavior and the social touch of NDDs. Given the lack of recent literature on tactile defensiveness, the current study provides a brief overview of the original contributions on this research topic in children with NDDs focusing attention on how this behavior has been considered over the years in the clinical setting.

## 1. Introduction

‘Noli me tangere’ or ‘Don’t touch me’ could be the silent cry that many children with neurodevelopmental disorders (NDDs) incessantly address to the world around them, with their nonverbal tactile defensive behaviors [1]. In this contest, the term ‘tactile defensiveness’ refers to an unusual avoidance–withdrawal response to non-threatening tactile stimuli or a hyperresponsivity to touch situations that most persons find non-noxious [2]. Such tactile defensiveness, in literature, has been generally felt in its epi-phenomenological dimension, as expression of an impairment in the processing of somatosensory information, assuming that the clinical sensory problems are due to perceptual deficits in processing tactile information [3]. One wonders, instead, if this cry cannot result from higher-level cognitive, emotional, and social factors and cannot be considered primarily as a compensatory attempt to communicate with the world, rather than the expression of damage to the system of basic recording and sensory modulation. In this perspective, the cognitive level is influenced in its modulation by autonomic and sensory systems [4,5,6,7].

Sensory processing abnormalities have been documented across all sensory modalities [8,9], and across all ages and levels of symptom severity [10] and often they may co-occur in the same individual [11]. However, while auditory and visual defensive behaviors have been sufficiently focused on for NDDs, perhaps due to the role of visual and auditory processing in verbal and non-verbal communication, the tactile defensive reactivity modality has been less studied, although abnormal touch responses are often described by parents in a wide variety of abnormal brain development (fragile X syndrome, autism spectrum disorder, attention deficit and hyperactivity disorders, cerebral palsy, early sensory deprivation) [1,8,12,13,14,15,16].

## 2. Aims and Methods

Tactile defensiveness had been an important study topic through the years. Given the lack of recent literature on this research topic, the aim of the current study is to collect an overview of the original articles reporting tactile defensiveness in children with NDDs and highlight how this behavior has been considered over the years in the clinical setting. A further aim of this study is to discuss the role of the tactile system in developmental socio-communicative dynamics. For this purpose, almost a hundred articles of the last ten years have been revised following the search criteria through keywords such as: neurodevelopmental disorders, intellectual developmental disorder, autism spectrum disorder, attention deficit and hyperactivity disorders, tactile defensiveness, tactile discrimination, social touch, sensory stimulation, sensory reactivity. Studies were identified through electronic database searching in Medline (Ovid, 1946 to present), PsycINFO (Ovid, 1806 to present), EMBASE (Ovid), and adapted for Scopus (Elsevier), ERIC (Proquest), PubMed, Web of Science (ISI), and Cochrane Library. The final database search was run on the 1 September 2019.

## 3. Discussion

According to the American Psychiatric Association [17], such neurodevelopmental disorders include intellectual disability (intellectual developmental disorder), communication disorders, autism spectrum disorder (ASD), attention deficit hyperactivity disorder, specific learning disorder, and motor disorders. Abnormal response to sensory stimulations represents a common behavior in NDDs. Kanner and Asperger included altered sensitivity to external sensory stimuli as a characteristic feature of ASD [18,19]. These sensory symptoms are so widespread that recently ‘hyper or hypo-reactivity to sensory input or unusual interests in sensory aspects of the environment’ has been added to the diagnostic criteria of ASD in the DSM-5 [17].

Three patterns of altered sensory reactivity have been identified [2]:Hyper-responsiveness to common environmental stimuli. Hyper-responsive individuals respond to low-intensity stimuli, showing a low response threshold for sensory events and a lack of habituation to continuous sensory stimulation. Therefore, they receive and respond to too many stimuli and avoid all the situations to which they attribute a negative affective value and that most people consider harmless.Hypo-responsiveness to common environmental stimuli. Hypo-responsive individuals respond to high-intensity stimuli, showing a high response threshold for sensory events, including high pain tolerance and a low responsiveness to sensory inputs.Sensory seeking behaviors. Seeker individuals perceive as pleasurable neutral stimuli and repeat a specific unusual stimulus situation that they consider particularly interesting and exciting.

Hyper-responsive children show discomfort for physical contact and for situations involving bodily contact (games, parties, social activities, supermarkets, cleaning). They dislike particular clothing items, certain textures, or particular materials and avoid tactile experiences [20,21]. Conversely, hypo-responsive children may react with pleasure to rough-and-tumble games, they can injure themselves, hit an obstacle and get bruises or constantly scratch a wound, and show indifference to pain, heat, or cold, while tactile seekers can seek out experiences of repetitive rubbing of certain textures or surfaces, or deep pressures such as intense hugs or squeezing [22]. Such anomalous tactile behaviors have generally been attributed to sensory dysfunctions [23], assuming that a hypo-responsive child does not react to a normal-level sensory input adequately because of hypo-sensitivity and a hyper-responsive child may overreact to a normal level stimulus because of hyper-sensitivity.

## 4. Tactile System: Developmental Primacy

For a broader understanding of the meaning of touch and tactile defensiveness, a focus on the role of touch in the human development and communication is needed. At birth, touch is more developed than the others sensory systems [24] and early bodily contact ‘skin to skin’ represents for the child the first modality of communication with the extra personal world and the primordial channel of access to information [3]. Somatosensory answers can even be elicited after the eighth week of gestation. After birth, the child’s survival is closely linked to sucking, that is to a reflex response to the tactile stimulation of the perioral area [25]. In the early years, mother–child communication, feeding, caring, and cuddling are almost exclusively mediated by a close body-touch contact. By tactile contact with the mother, the child feels and communicates emotions and creates maternal bonds, learning to know the mother, to feel her way of being a mother, her mood, emotions, feelings, affection, anxieties, fears, and uncertainties. Being cared, touched, caressed, and lovingly tickled conveys to the child affection, reassurance, well-being, relaxation, and a secure attachment critical for survival [26,27,28] and for the development of the child’s basic feeling of trust in himself, in life, and in the environment [29]. According to Bowlby [30,31], feeling the physical closeness and contact to the mother is the main signal for children to be safe and protected. Interpersonal touch in early life is strongly associated with the development of secure attachment [32,33] and basic family bonds [34] and influence neural and behavioral social development [35]. In Harlow’s classic experiment, the infant monkey isolated from the mother seeks ‘contact comfort’ and clings to the surrogate mother of soft material rather than the other made of wire, regardless of which one could provide food.

Harlow’s seminal search was the experimental model for several animal studies on the role of contact with and early proximity to caregivers in other mammals [36]. In rodent studies, some rat pups were briefly removed from their mother and then returned. Those pups who received more tactile stimulations, such as licking and grooming, on their return, showed greater resistance to stress. Conversely, those who had received less grooming and licking showed greater reactivity to stress. Moreover, their offspring were more stress reactive [36].

The mother–child attachment is deeper in human newborns receiving skin-to-skin care [37], with positive effects on heart rate, respiratory rate, and oxygen saturation [38]. Similarly, preterm infants who receive a slow and caressing touch gain weight and spend less time in the hospital than controls [39]. A richer vocalization and smiling were found in infants who experienced systematically affectionate maternal touch. The same children showed better social communication later in life [40,41,42].

Conversely, poor tactile interactions early in infancy may result in aberrant repetitive behaviors, echolalia, stereotypies, and can have a serious negative impact on the child’s psychomotor and even physical development [43]. Probably, the quality and intensity of this first modality influences future relationships and affects the development of non-verbal and verbal communication for the rest of life. Indeed, even after the appearance of verbal communication, physical contact and tactile input continue to be a primary mode of communication that improves verbal communication itself, even in the adult stage [44]. Therefore, early tactile communication can be seen as a precursor of verbal communication [45].

On the other hand, considering an evolutionary framework, the non-verbal tactile communication, like grooming behaviors, comes before verbal communication [46]. Monkeys, for example, practice grooming for much longer than is hygienically required [45,47]. Furthermore, grooming behavior does not only have a purely hygienic purpose, but can also have an important emotional and social goal. Time of grooming takes much longer as the social group grows larger. It is therefore likely that the most important role of grooming contact is to promote or reinforce positive relationships within the group and keep it cohesive. It is also probable that when hominids organize themselves into larger groups and seek food in larger areas, a verbal communication modality becomes necessary, in addition to the contact mode and tactile stimulus. Therefore, it can be hypothesized that in the ontogenesis of cuddling and skin-to-skin contact between preverbal children and their mother, the phylogenetic role of grooming, as a precursor behavior of verbal communication, is summarized [45]. Touch therefore is crucial for social development in early childhood [40,42] and represents a primordial access channel for early interpersonal relations [28,40,42] on which rests the future affective and relational development.

## 5. Tactile System: Discriminative and Affective Dimension

Tactile information-processing starts in the sensory receptors of the skin and, through the spinal and thalamic pathways, is relayed to the cortical sensory areas. The primary somatosensory cortex processes elementary tactile information in a somatotopic organization, while the associative cortex processes and integrates the individual basic information into a significant higher-level perceptual act [48]. Tactile sensory modality allows us to manipulate objects and explore their haptic features as shape, textures, thickness, roughness, softness, fragility, and consistency, allowing us to perceive tactile experiences as pleasant or unpleasant. Moreover, interpersonal touch can promote communication with each other by a range of tactile social interactions.

In human neurophysiology, the tactile system includes two parallel and functionally different peripheral and cortical pathways for discriminative and affective touch [48,49]. Discriminative touch provides, in an exteroceptive fashion, haptic features of an object. It is activated by any type of touch on the skin and primarily mediated by A-beta and A-delta fibers, a class of fast-conducting, myelinated, large-diameter peripheral nerves distributed in the hairless, glabrous skin of the palm and projecting to the discriminative-cognitive system of the primary and secondary somatosensory cortexes [48]. Affective touch elicits hedonic or emotional responses, supporting the subjective experience of affiliative and emotional somatic pleasure of touch [50]. It is activated selectively by caress-like gentle touch and mediated by C-tactile (CT) afferents, a class of slow-conducting, unmyelinated, small-diameter, low-threshold, mechanoreceptive peripheral fibers, distributed primarily in the hairy skin and in the face [49] and projecting mainly towards the emotional, affect-related cortical regions [1,51,52,53,54,55,56] such as the anterior cingulate, insular, and orbitofrontal areas [49,50,57,58,59], the temporoparietal junction, and the superior temporal sulcus [60,61].

In sum, considering its anatomical and physiological properties, such as fiber class, slow conduction, as well as limbic-emotional areas of cortical projection, hedonic, and affective nature, the CT-spinothalamic system share more characteristics with interoceptive modalities generating autonomic homeostatic emotional and behavioral responses [62], whereas rapid and accurate A-beta fibers reflect the external world. Indeed, although interoceptive CT information arises from the external surface of the organism, CT afferents follow a similar route to the brain projection areas as visceral thin fibers, creating an area of overlap between visceral afferents and cutaneous afferents. In addition, as CT projects to the interoceptive cortex in the posterior insula, it contributes to the subjective awareness of the body’s state and to maintaining homeostatic balance [63]. Therefore, whereas discriminative touch encodes the presence on the skin of a stimulus and its objective tactile characteristics, the affective touch encodes the emotional, affective, relational, and social features of a tactile stimulation and its relevance in affiliative context. Taken together, these findings provide a relevant support for an affective touch hypothesis [48].

However, CT afferents may be somehow considered within a large interoceptive system for emotional aspects of tactile perception, monitoring the physiological and chemical variables supporting limbic–emotional, autonomous, hormonal, and behavioral responses [62] to tactile contact with con-specifics [64]. It is interesting to highlight that people with autism spectrum disorder and tactile defensiveness exhibited, in functional magnetic resonance imaging, reduced activity in response to CT-targeted versus non-CT-targeted touch in brain areas involved in social–emotional information processing, suggesting atypical social brain hypoactivation. Whereas they showed an enhanced response to non-CT-targeted versus CT-targeted touch in the primary unimodal somatosensory cortex, suggesting atypical sensory cortical hyper-reactivity [54,65,66].

## 6. Social Touch and Social Communication

Recent findings suggest that stimulation of C-tactile afferents correlates with activation of regions associated with social cognition [67,68,69,70], supporting the hypothesis that skin is a ‘social organ’ and that C-tactile afferents may be a part of a social communication system [48,54,58,60,71,72,73].

Several experimental studies have shown that interpersonal touch has played an important role as a communication channel since the first social interactions. Even the most simple and immediate social touches, like a caress, a handshake, a pat on the shoulder, or a push can communicate significant positive or negative emotional experiences and improve the meaning of other forms of verbal and non-verbal communication. Eye contact with other people may have a different meaning depending on whether or not we touch them simultaneously. In a classic experiment, library clerks were asked to return the library’s card to the students and, while they were doing it, to get their hands directly on the palms only of some student. The students who, without even realizing it, had been ‘accidentally’ touched by the clerks gave more positive evaluations about the library [74]. Similarly, in a store, customers tend to respond more positively to a request for tasting and buying and are more likely to agree to participate in interviews when they are touched by an experimenter who acts as a store clerk than when nobody touches them [75]. Likewise, among the students who had been touched briefly by the teacher during a statistical exercise, the highest number of those who volunteered to demonstrate the solution on the board were registered [76]. Other research has shown that interpersonal touch can be successfully used to share emotional aspects of communication. Participants were asked to identify emotions from the experience of being touched on the arm by another unknown participant. The latter was asked to touch the bare arm of a subject from the elbow to the end of the hand to signal specific emotions. The results showed that the participants were able to decode emotions to an extent comparable to the success rates of transmission and decoding of facial displays and voice communication [46].

This kind of research appears to be sufficiently in line with the data of neurophysiological studies. Touch should not only be seen as a cutaneous modification that gives us discriminative haptic information about the external world. It is also a communication channel that enriches interpersonal relationships from infancy onward and allows us to improve social cognition. In a recent study, Aguirre and colleagues [77] have shown in normal 9-month-old infants, stroked to the legs with a brush at a different speed by either an unfamiliar experimenter or a caregiver, that the child’s heart rate decreased more, showing greater relaxation, when strokes were given by caregivers rather than by strangers. Moreover, this effect was found only for tactile stimulation whose velocity was maximal mean firing rates in afferent C-tactile fibers. Therefore, already in the first year of life, tactile stimulation is not a purely mechanical event that affects the skin but it expresses a pleasant or less pleasant relationship. Similar data had been found in a previous study on two-month-old children in which stroke of intermediate velocity (3 cm/s) activated brain areas that were affective-related, such as the temporal and insular cortex, more than faster strokes [78].

Therefore, interpersonal touch is strongly influenced by its social properties and by specific channels that can contribute to social cognition, such as CT and the projection areas of the brain, playing a social and communicative function. From childhood onwards, then, the discriminative and affective components of touch interact with a sensitivity to the identity of the source of touch. Finally, touch plays a key role in building a representation of the body self which in turn is crucial to stand out from others, engage in social interaction, and predict and interpret the behavior of others.

In this context, it may be exciting to note Adolphs and colleagues’ research [69,79] showing that bilateral amygdala damage compromises interpersonal space and the degree of close physical proximity. The patient ‘without amygdala’ does not claim any discomfort at close interpersonal distances even when standing ‘nose to nose’ with the experimenter [79]. Probably, according to Adolphs, she cannot detect the socially and emotionally salient aspects of the situation and the feelings related to physical distance.

Social and interpersonal touch, as a simple tap, protracted hug, or dynamic caress, may be regarded as an important category of the affective touch and it may be a crucial mediator of affiliative behavior and communication, and intersubjective representations of others’ sensory, emotional, and mental states. Therefore, it can promote social bonds by a range of tactile social interactions [35]. There is growing neurobiological evidence that the gratifying meaning of physical contact in social interactions arises from a mechanism, mediated by C-tactile inputs, which promotes contact in specific contexts [48]. Touch is the earliest sensory modality to develop [24,80], and the unmyelinated system may already be functional at birth, and two months after birth may be functional in an adult-like manner [78,81]. Moreover, while discriminative tactile abilities with age may decrease, perceived pleasantness of CT-targeted touch continues to increase until old age [48].

Therefore, the tactile system supports both a discriminating touch and an affective touch within a complex functional system originating in the skin, as the sensory access channel, and ending in the primary somatosensory brain areas and in the associative areas of higher polymodal integration and emotional processing, where the basic tactile stimulus becomes cognition, affection, and feeling [82]. In this perspective, the abnormal development in any element of the somatosensory functional system, both low-order or high-order, may involve the functionality of the entire functional system of tactile processing and it promotes behavioral responses of tactile defensiveness.

## 7. Tactile Threshold or Emotional and Social Impairment

Generally, in literature, tactile defensiveness was associated with developmental impairment of the tactile perceptive threshold [83]. However, it should be noted that the data from the literature mainly result from subjective reports of parents, caregivers, and even the high-functioning subjects [22,83]. Additionally, these empirical findings focused on lower-order dysfunctions of the somatosensory system, neglecting higher-cognitive order, as well as impaired emotional processing and social communication. However, in clinical settings, self-rating scales, reported by parents, caregivers, or by subjects, continue to be used, although semi-structured interviews are more valid to reduce the subjective bias of the reports. In the experimental field, the most accurate measures prove the quantitative methods of psychophysics to study sensation and perception. Recently, studies using objective measures of sensory processing and applying psychophysical assessment methods provide more specific evidence for potential mechanisms underlying sensory impairment [1,3,83].

On such psychophysical measures, several studies about tactile defensiveness reported both in adults and in children with NDDs thresholds comparable to controls and, in addition, significant correlations between tactile and affective items on parent questionnaires [3,66,83,84,85].

For example, Guclu, et al., comparing boys affected by ASD with typically developing controls in tactile detection threshold in two different experimental conditions, did not find differences between groups in tactile thresholds [66]. Moreover, they found a correlation between tactile and emotional items of the Touch Inventory for Elementary-School-Aged Children and Sensory Profile. The authors hypothesized that the abnormal tactile sensitivity in ASD could be related to emotional impairments and could be more present with concomitant emotional problems.

Previously, O’Riordan and Passetti [86], studying the performance of children with and without ASD on tactile discrimination tasks, identified a comparable tactile discrimination in ASD with respect to controls. Similarly, comparing tactile sensation, Cascio, et al. [65] found that ASD adults and controls displayed similar thresholds for detecting light touch and innocuous sensations of warmth and cool, and provided similar hedonic ratings of the pleasantness of textures suggesting that tactile defensiveness in ASD may be at least be partially modulated by affective neural systems of touch as opposed to discriminative touch pathways.

Interestingly, a study by Blakemore and colleagues [84] investigated in people with autism whether hypersensitivity would be found within certain tactile stimuli and not others. They examined in individuals with Asperger syndrome and a normal control sensitivity to vibrotactile stimuli at two different frequencies (30 and 200 Hz), given that high-frequency vibration (200 Hz) stimulates Pacinian corpuscles and activates low-threshold fast adapting mechanoreceptors (FAII fibers), and lower-frequency vibration (30 Hz) stimulates Meissner corpuscles and activates slowly adapting-low-threshold mechanoreceptors (SAI fibers) well adapted for high-resolution discrimination of shape and texture. The study confirmed a slight tactile hypersensitivity in Asperger people but only for high-frequency stimuli (200 Hz). In a second study, the authors found that the perceptual consequences of self-produced touch are attenuated in the normal way in people with Asperger syndrome, suggesting that the neural changes underlying tactile sensory problems do not affect absolute thresholds, but modify intensity discrimination or magnitude estimation. Moreover, they suggest that hypersensitivity occurs at some as yet unidentified neural level.

The hypersensitivity to touch may be due to an abnormal processing of touch in one or more components of the tactile system. Additionally, our measure of tactile threshold pertains to the discriminative rather than the affective pathway. Therefore, hypersensitivity to suprathreshold tactile stimuli it could be a response to a particularly high stimulus rather than a dislike of a normal stimulus. Individuals with autism may show a normal threshold in detecting a simple stimulus, but elevated thresholds when detecting a complex, second-order stimulus. The hyper-responsiveness, then, expresses an impairment in processing rapidly changing, dynamic stimuli as well as an increased sensitivity to second order, complex stimuli that require additional integration of information.

However, psychophysical studies of discriminative touch processing have yielded inconsistent results. It would seem that the emotional aspects of touch are more consistently affected in ASD children [65,66]. Several studies are increasingly highlighting the emotional aspects of tactile stimulation and the emotional–relational meaning of tactile defensiveness.

Overcoming the classic relationship between tactile defensiveness and low or high threshold levels, perhaps the tactile defensiveness could be rethought as the phenomenological expression of the emotional and social communication problems of subjects with NDDs. Maintaining a comfortable physical proximity with others on the basis of feelings and personal comfort is the expression of a social and emotional ability that allows us to correctly evaluate interpersonal distance as socially and emotionally significant. It would seem that NDDs subjects cannot modulate the emotional dimension expressed by the physical distance with others. The NDDs subjects come too close or too distant, showing in either case that they cannot adequately modulate distances and, above all, their relational and emotional meaning.

## 8. Conclusions

In conclusion, the current overview has collected suggestions and hypotheses to facilitate the carrying out of further research and more evidence on the complex reality of the sense of touch and tactile defensiveness in NDDs. In literature, tactile defensiveness has received much less attention than defenses behavior in visual and auditory modality. Probably for this reason many outstanding questions are still open and many areas remain poorly explored both in touch sense and in tactile defensiveness areas.

A first crucial area concerns the matrix of tactile defensiveness. Future clinical and experimental studies will have to provide evidence to understand if the tactile defense responses can arise mainly from an abnormal sensory–tactile threshold in the NDDs, as in the literature’s prevailing interpretation, or if, instead, they can be thought as a behavioral manifestation of a more general disorder of social communication and interpersonal relationships.

Related to that, it becomes interesting to consider whether abnormal tactile reactions can be thought of as the use of a primordial language and communication modalities in individuals with a broad relationship and communication disorder.

Moreover, it could be interesting to study whether the hypo- or the hyper-tactile responses in NDDs can be somehow associated with the difficulties to regulate close physical proximity and appropriate social distance and whether both tactile defensiveness and the difficulties of modulating the physical distance between people can be thought of more broadly as a signal of a single disorder of the modulation of emotions related to interpersonal relationships.

Hopefully, future clinical and experimental research can also improve knowledge about the role of the sense of touch in general in interpersonal relationships in both typical development and neurodevelopmental disorders.

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
