# Peer review of "Noli Me Tangere: Social Touch, Tactile Defensiveness, and Communication in Neurodevelopmental Disorders"

_brainsci, 2019, doi:10.3390/brainsci9120368_

Round 1

Reviewer 1 Report

The manuscript is a review of the literature on a theme of the tact modality in the neurodevelopmental disorder (NDD). This review brings essential knowledge about this modality in the field of NDD, especially regarding the tactile defensiveness.

Some point must be improved:   

Major point:

The first section of the discussion on “Discriminative and affective touch” needs to be improved to better understand the link between touch, interoception and emotions. Sometimes ideas are not clearly presented and it becomes difficult to follow further discussion on the involvement of the autonomic nervous system in the disturbance of tactile reactivity would be appreciated

Minor point:

In the introduction, bibliographic references are missing (especially at the beginning) In the methodology, a quantification of the number of works analyzed should be indicated In the methodology, the process that led to the organization of this review work should be clarified In the discussion, p2, lines 68 to 77 the line spacing is not homogeneous with the rest of the manuscript In the discussion, p4, lines 155 to 156, lines 161 to 163, and lines 175 to 179, the link between touch and intercept must be supported. I do not agree with the fact that touch is considered as an interoceptive system.Interoceptive changes are the consequence of tactile stimulation (exteroceptive). In the discussion, p4, lines 172 to 179, the discussion on C tactile fiber must be in the same paragraph In the discussion, p5, line 185, the transition to the paragraph on the amygdala must be worked In the discussion, p5, line 210, “adults and in children” with NDD ? In the discussion, p5, lines 213 to 214, “it may arise from cognitive and emotional mechanisms, rather than from differences in 213 physiological response of the somatosensory system”…. it is not the only possible interpretation In the discussion, p6, lines 242 to 246, should the authors include a brief paragraph on the physio-anatomy of tactile modality at the beginning of the discussion rather than having a paragraph like this here? In the discussion, p6, lines 268 to 278, this paragraph is redundant with that p5 lines 185 to 192

Author Response

The manuscript is a review of the literature on a theme of the tact modality in the neurodevelopmental disorder (NDD). This review brings essential knowledge about this modality in the field of NDD, especially regarding the tactile defensiveness.

Some point must be improved:   

Major point:

The first section of the discussion on “Discriminative and affective touch”needs to be improved to better understand the link between touch, interoception and emotions. Sometimes ideas are not clearly presented and it becomes difficult to follow further discussion on the involvement of the autonomic nervous system in the disturbance of tactile reactivity would be appreciated.

We thank the reviewer for this suggestion which has allowed us to substantially improve the manuscript. That section of the discussion is no longer entitled “Discriminative and affective touch” but it has been replaced withTactile System: discriminative and affective dimension”. It is no longer in the first section of the discussion and has been substantially changed in order to make the links between touch, interoception and emotions clearer. We hope that in this revised version of the manuscript the ideas are now more clearly presented and it is easier to follow the discussion on the involvement of the autonomic nervous system in the tactile reactivity disorder (see page 4).

Minor point:

In the introduction, bibliographic references are missing (especially at the beginning)

We apologize for this oversight. We added missing references, especially at the beginning, following the reviewer’s suggestion.

In the methodology, a quantification of the number of works analyzed should be indicated In the methodology, the process that led to the organization of this review work should be clarified.

We thank the reviewer for this suggestion which has allowed us to clarify the selection criteria for the articles collected, and to quantify the number of works analyzed (see page 2 lines 70-78).

In the discussion, p2, lines 68 to 77 the line spacing is not homogeneous with the rest of the manuscript.

We apologize for this. We have made the line spacing homogeneous with the rest of the manuscript following the reviewer’s suggestion (see page 2 lines 88-98).

In the discussion, p4, lines 155 to 156, lines 161 to 163, and lines 175 to 179, the link between touch and intercept must be supported. I do not agree with the fact that touch is considered as an interoceptive system. Interoceptive changes are the consequence of tactile stimulation (exteroceptive).

We thank the reviewer for this useful suggestion. In this revised version of the manuscript in the discussion session, we have deepened the link between touch and interoception (the lines mentioned by the reviewer are no longer present, see the rewritten paragraph in page 4 from line 418 to 678).

In the discussion, p4, lines 172 to 179, the discussion on C tactile fiber must be in the same paragraph

We thank the reviewer for this suggestion. The paragraph on C tactile fiber is no longer present in that form in this revised version of the manuscript.

In the discussion, p5, line 185, the transition to the paragraph on the amygdala must be worked.

We thank the reviewer for this comment. We now added a new paragraph entitled “Social Touch and Social Communication”, where the amygdala paragraph was moved (see pages 5 and 6, lines 679-749)

In the discussion, p5, line 210, “adults and in children” with NDD ?

We thank the reviewer for this comment. We now added “with NDDs” (see page 6 line 763)

In the discussion, p5, lines 213 to 214, “it may arise from cognitive and emotional mechanisms, rather than from differences in 213 physiological response of the somatosensory system”…. it is not the only possible interpretation.

We agree with the reviewer. In this revised version of the manuscript we have explained this concept better by clarifying that it is not the only possible interpretation (see page 6)

In the discussion, p6, lines 242 to 246, should the authors include a brief paragraph on the physio-anatomy of tactile modality at the beginning of the discussion rather than having a paragraph like this here?

We thank the reviewer for this suggestion. We have totally reorganized the discussion in this revised version of the manuscript.

In the discussion, p6, lines 268 to 278, this paragraph is redundant with that p5 lines 185 to 192.

We thank the reviewer for this comment. We apologize for this oversight, in fact we found a repetition, which we have now deleted.

Reviewer 2 Report

Smirni and colleagues present a well written and comprehensive review on a timely subject, tactile defensiveness in neurodevelopmental disorders (NDDs). The review sheds light on a subject that's not often discussed or recognized in NDDs. 

Author Response

We thank the reviewer for these positive comments.

As requested we checked the spelling in this revised version of the article.

Reviewer 3 Report

The authors present a timely perspective on tactile defensiveness in neurodevelopmental disorders (NDDs). This piece primarily focuses on the role of the tactile system in social functioning and communication. Despite the brevity of the piece, the scope of the article is broad, encompassing both behavioral and physiological aspects of tactile discrimination in typical and atypical development. Though the piece serves as an excellent introduction, limitations in the organization and methodology curtail the authors potential impact.

Comments to the Authors:

This interesting manuscript would benefit from a thorough reorganization to more clearly support and illustrate the major points of discussion and eliminate redundancy.

First, the authors should consider carefully if the stated goal of this perspective should be shifted to discuss the role of the tactile system in socio-communicative dynamics (or tactile discrimination) in a more broad sense (which is the primary content of the present manuscript) rather than tactile defensiveness, which has a much more narrow scope. Much of the content (i.e. lines 100-199) are primarily discussing the role of the tactile system in development and communication rather than tactile defensiveness.

In terms of methodology, critical details are omitted such as key words, search portals, and the “time stamped” results of these searches. Despite the relatively informal nature of a perspectives piece, addressing these details will benefit future papers on this topic.

In terms of organization, the reader would benefit from the authors more clearly defining tactile defensiveness prior to expounding on sensory abnormalities in NDDs. These two concepts are somewhat mingled throughout the paper. This is only my opinion, but gathering all discussion regarding the normal development, physiology, and components of the tactile system into a single section may clarify future sections. It seems as though the authors argument that tactile defensiveness encompasses emotional and cognitive circuits would be strengthened by greater emphasis on central circuits (including links to the motor system which is not well represented in the present manuscript) in addition to describing peripheral components. In a revised format, potentially greater space could be dedicated to expounding on new experimental research establishing a link between touch and social cognition should be further expounded (as started in line 184).

Consider areas of redundancy in the paper. For example, both lines 188-190 and 272-274 describe reference the same experiment (81) regarding the relationship of the amygdala and personal space. The authors first would have to establish through argument that this phenomenon represents “tactile defensiveness” and can be generalized to NDDs, because in the current revision these only seem peripherally related.

Lines 259-267: The authors make several strongly worded statements (i.e. “it would seem that NDDs subjects cannot modulate the emotional dimension expressed by the physical distance with other”, however, there is inadequate supportive evidence referenced and if speculative, should be stated as such. Especially given clinical observations that many children with NDDs may be sensory and tactile seeking (i.e. heavy pressure, tight squeezes from a caregiver).

It seems given the paucity of experimental research in humans regarding tactile defensiveness specifically, the authors should consider briefly expanding the discussion to discuss the phenomena as observed in other mammals. Furthermore, I would challenge the authors to more clearly describe in the Conclusion specific gaps in the literature and potential approaches to answer key outstanding questions in the field. 

In conclusion, I am mixed regarding my recommendation. Though I am pleased to see interest in this rare and relevant topic, I would recommend  substantial revisions to the organization and methodology.

Minor Revisions:

Line 50: Methods. Please add key words, manuscript indexer, and date of search for context.

Line 201-214: What is the gold standard objective measure of “tactile defensiveness”?

Author Response

As requested we checked the spelling in this revised version of the article.

Comments to the Authors:

This interesting manuscript would benefit from a thorough reorganization to more clearly support and illustrate the major points of discussion and eliminate redundancy.

Thank you for defining our manuscript interesting. We thank the reviewer for the useful suggestions that allowed us to improve our work and make it more readable and clearer. Our revised version of the manuscript has been substantially modified, reorganized and redundancies have been eliminated.

First, the authors should consider carefully if the stated goal of this perspective should be shifted to discuss the role of the tactile system in socio-communicative dynamics (or tactile discrimination) in a more broad sense (which is the primary content of the present manuscript) rather than tactile defensiveness, which has a much more narrow scope. Much of the content (i.e. lines 100-199) are primarily discussing the role of the tactile system in development and communication rather than tactile defensiveness.

We thank the reviewer for this helpful suggestion. We agree with the reviewer and following his/her suggestion we decided to better specify and clarify the aims of this study (see page 2 lines 66-71).

In terms of methodology, critical details are omitted such as key words, search portals, and the “time stamped” results of these searches. Despite the relatively informal nature of a perspectives piece, addressing these details will benefit future papers on this topic.

We thank the reviewer for this suggestion. In this revised version of the manuscript we added the missing information following the reviewer's instructions (see page 2 lines 71-78).

In terms of organization, the reader would benefit from the authors more clearly defining tactile defensiveness prior to expounding on sensory abnormalities in NDDs. These two concepts are somewhat mingled throughout the paper.

We agree with the reviewer and, in terms of organization, we have included a clear definition of tactile defensiveness before explaining sensory abnormalities in NDDs (see page 1 and start of 2 line 39-41). We hope this revised organization has improved the manuscript in terms of readability. We also hope to have disambiguated the two concepts throughout the paper.

This is only my opinion, but gathering all discussion regarding the normal development, physiology, and components of the tactile system into a single section may clarify future sections.

We thank the reviewer for this suggestion. We followed his/her directions and brought together all the discussion related to the normal development, physiology and components of the tactile system in a single section, considering that this may clarify future sections (see page 3 lines 203-394).

It seems as though the authors argument that tactile defensiveness encompasses emotional and cognitive circuits would be strengthened by greater emphasis on central circuits (including links to the motor system which is not well represented in the present manuscript) in addition to describing peripheral components. In a revised format, potentially greater space could be dedicated to expounding on new experimental research establishing a link between touch and social cognition should be further expounded (as started in line 184).

We thank the reviewer for this suggestion. In this new version of the manuscript a greater space has been dedicated to experimental research establishing a link between touch and social cognition  (see page5 the new paragraph entitled Social Touch and Social Communication)

Consider areas of redundancy in the paper. For example, both lines 188-190 and 272-274 describe reference the same experiment (81) regarding the relationship of the amygdala and personal space. The authors first would have to establish through argument that this phenomenon represents “tactile defensiveness” and can be generalized to NDDs, because in the current revision these only seem peripherally related.

We apologize for this oversight. We have now deleted the redundancy and we have better argued that the phenomenon observed in the described experiment could be generalized to NDDs (see page 6).

Lines 259-267: The authors make several strongly worded statements (i.e. “it would seem that NDDs subjects cannot modulate the emotional dimension expressed by the physical distance with other”, however, there is inadequate supportive evidence referenced and if speculative, should be stated as such. Especially given clinical observations that many children with NDDs may be sensory and tactile seeking (i.e. heavy pressure, tight squeezes from a caregiver).

We apologize for this. In this revised version of the manuscript we tried to have more cautious statements where there was inadequate supportive evidence.

It seems given the paucity of experimental research in humans regarding tactile defensiveness specifically, the authors should consider briefly expanding the discussion to discuss the phenomena as observed in other mammals.

Following the reviewer's suggestion we have expanded the discussion also on tactile defensiveness observed in other mammals (see page 3, lines 220-228).

Furthermore, I would challenge the authors to more clearly describe in the Conclusion specific gaps in the literature and potential approaches to answer key outstanding questions in the field. 

Thank you for this useful suggestion that allowed us to highlight, in the conclusion section, which further research needs to be considered on this topic and which questions still remain unanswered.

In conclusion, I am mixed regarding my recommendation. Though I am pleased to see interest in this rare and relevant topic, I would recommend  substantial revisions to the organization and methodology.

Thank you for considering our manuscript for publication and for your encouragement to submit a revised version.

Minor Revisions:

Line 50: Methods. Please add key words, manuscript indexer, and date of search for context.

We have greatly expanded the missing information.

Line 201-214: What is the gold standard objective measure of “tactile defensiveness”?

We have clarified the definition and objective measures of “tactile defensiveness” (see page 6, lines 756-761)